# MOB (Mps one Binder) Proteins in the Hippo Pathway and Cancer

**DOI:** 10.3390/cells8060569

**Published:** 2019-06-10

**Authors:** Ramazan Gundogdu, Alexander Hergovich

**Affiliations:** 1Vocational School of Health Services, Bingol University, 12000 Bingol, Turkey; rgundogdu@bingol.edu.tr; 2UCL Cancer Institute, University College London, WC1E 6BT London, UK

**Keywords:** Mps one binder, Hippo pathway, protein kinase, signal transduction, phosphorylation, protein-protein interactions, structure biology, STK38, NDR, LATS, MST, STRIPAK

## Abstract

The family of MOBs (monopolar spindle-one-binder proteins) is highly conserved in the eukaryotic kingdom. MOBs represent globular scaffold proteins without any known enzymatic activities. They can act as signal transducers in essential intracellular pathways. MOBs have diverse cancer-associated cellular functions through regulatory interactions with members of the NDR/LATS kinase family. By forming additional complexes with serine/threonine protein kinases of the germinal centre kinase families, other enzymes and scaffolding factors, MOBs appear to be linked to an even broader disease spectrum. Here, we review our current understanding of this emerging protein family, with emphases on post-translational modifications, protein-protein interactions, and cellular processes that are possibly linked to cancer and other diseases. In particular, we summarise the roles of MOBs as core components of the Hippo tissue growth and regeneration pathway.

## 1. Introduction

The family of MOBs (monopolar spindle-one-binder proteins) is highly conserved in eukaryotes [1,2,3,4]. To our knowledge, at least two different MOBs have been found in every eukaryote analysed so far. For example, in unicellular organisms such as yeast the MOB proteins Mob1p and Mob2p are expressed by two independent genes [5]. In multicellular organisms such as flies, at least four different MOBs, termed dMOBs (Drosophila MOB proteins), have been reported [4]. In humans, as many as seven different MOB proteins, termed hMOBs (human MOB proteins), are encoded by different gene loci. Since each hMOB has been given several names over the years, we have simplified the nomenclature of hMOBs as follows [1]: hMOB1A (UniProtKB: Q9H8S9; also termed MOBKL1B, MOBK1B, Mob1a, MOB1α, Mob4b, MATS1 and C2orf6), hMOB1B (UniProtKB: Q7L9L4; also termed MOBKL1A, Mob1b, Mob4a and MATS2), hMOB2 (UniProtKB: Q70IA6; also termed MOBKL2, Mob2 and HCCA2), hMOB3A (UniProtKB: Q96BX8; also termed MOBKL2A, Mob3A, MOB-LAK, MOB2A, Mob1C), hMOB3B (UniProtKB: Q86TA1; MOBKL2B, Mob3B, MOB2B, Mob1D and C9orf35), hMOB3C (UniProtKB: Q70IA8; also termed MOBKL2C, Mob3C, MOB2C and Mob1E) and hMOB4 (UniProtKB: Q9Y3A3; also termed MOBKL3, Class II mMOB1, MOB3, Phocein/PHOCN, PREI3, 2C4D, and CGI-95). Given the very high identity between hMOB1A and hMOB1B (Figure 1), we refer to hMOB1A/B also as hMOB1 in this review.

In comparing the sequence identities of MOB proteins expressed in fly and human cells (Figure 1), it becomes apparent that hMOB1 is highly conserved in flies. dMOB1 is 85% identical to hMOB1 and functionally conserved, since exogenous expression of hMOB1A rescues the phenotypes associated with dMOB1 loss-of-function [6,7]. Among other MOBs, hMOB1 is the most closely related to hMOB3A/B/C and dMOB3 (Figure 1). hMOB2 is not as well conserved as hMOB1, since hMOB2 aligns similarly with dMOB1 and dMOB2. hMOB3A is very similar to hMOB3B and hMOB3C, being also 64% identical to dMOB3. hMOB4 is 80% identical to dMOB4, and hMOB4 and dMOB4 do not show any significant identity overlaps with other MOBs (Figure 1). A phylogenetic tree analysis of the dMOBs and hMOBs showed that hMOB1A, hMOB1B and dMOB1 cluster together into one subgroup (Figure 2). Likewise, hMOB2 and dMOB2, as well as hMOB3A, hMOB3B, hMOB3C and dMOB3 form separate subgroups, respectively. It is noteworthy that hMOB4 and dMOB4 also form a subgroup of their own, representing the most distant subgroup of MOBs (Figure 2). However, in spite of these striking sequence similarities, it is currently unknown whether loss-of-function of dMOB2, dMOB3 or dMOB4 can be functionally compensated by the expression of the corresponding hMOB family member. Nevertheless, research covering the past two decades has uncovered important aspects of MOBs that seem to be universally valid for all members of the MOB family. In summary, it was found that MOBs mainly function as intracellular co-regulatory proteins. In particular MOBs can directly bind to protein kinases, and thereby influence the activities of their binding partners. As such, before discussing MOBs in flies and mammals, we will provide a brief introductory background on MOBs studied in unicellular organisms such as yeast.

More than two decades ago Mob1p, the first MOB (monopolar spindle-one-binder) protein, was described in budding yeast [1,5,8]. It turned out that Mob1p is a central component of the mitotic exit network (MEN) and the septation initiation network (SIN) of budding and fission yeast, respectively (summarised in refs. [8,9,10,11,12]). Mechanistically, budding yeast Mob1p in complex with the Dbf2p protein kinase (a budding yeast counterpart of mammalian NDR/LATS kinases [13]) regulates the release of phosphatase Cdc14p to promote exit from mitosis through a series of dephosphorylation events. Similarly, fission yeast Mob1p bound to Sid2p (a fission yeast counterpart of mammalian NDR/LATS kinases) controls the Clp1p phosphatase (the equivalent of Cdc14p) to support the SIN. Mob2p, another member of the MOB family [1], was uncovered together with Mob1p in budding yeast [5]. Subsequent research revealed that Mob2p coordinates morphogenesis networks in budding and fission yeast [1,13], as well as fungal species [14]. In budding yeast, Mob2p in complex with Cbk1p (the second NDR/LATS kinase expressed in budding yeast [13]) controls the RAM (regulation of Ace2p activity and cellular morphogenesis) signalling network. In fission yeast, Mob2p associated with Orb6p (the second NDR/LATS kinase in fission yeast) coordinates a similar morphogenesis network.

In summary, yeast cells express two different MOB proteins, Mob1p and Mob2p. While Mob1p controls mitotic exit, Mob2p is a regulator of cell morphogenesis and polarized growth. Mob1p and Mob2p function in complex with different members of the NDR/LATS kinase family. Depending on the yeast species Mob1p binds to Dbf2p or Sid2p whereas Mob2p associates with Cbk1p or Orb6p, respectively. Consequently, in yeast Mob1p and Mob2p act as part of independent and non-interchangeable complexes containing different NDR/LATS kinases in order to regulate diverse cellular processes (please consult refs. [1,13] for a more detailed discussion of these points).

## 2. An Overview of MOBs in *Drosophila melanogaster*

Since 2005, MOBs have been studied in multicellular organisms. In contrast to yeast, fly cells encode four different MOB proteins [4]. Lai and colleagues found that dMOB1 (also termed Mats for MOB as tumour suppressor) is an essential regulator of cell proliferation and cell death in fruit flies [7]. Follow up studies established dMOB1 as a core component of the Hippo tissue growth control pathway (for more details see subchapter 4 of this review). Briefly, as a core component of the Hippo pathway, dMOB1 acts as a co-activator of the Warts protein kinase [15,16,17], one of two NDR/LATS kinases in *D. melanogaster* [13]. Intriguingly, dMOB1 also genetically interacts with Tricornered (Trc) [4], the second NDR/LATS kinase in flies [13]. Thus, dMOB1 does not bind specifically to a single NDR/LATS kinase, as observed for yeast Mob1p [1]. In addition, dMOB1 can form a complex with the Hippo (Hpo) protein kinase [18], with Hpo being able to phosphorylate dMOB1 [18] as well as Warts and Trc [19]. The importance of these protein-protein interactions and phosphorylation events is discussed in subchapters 4, 5 and 6. Noteworthy, dMOB1 can further play a role in mitosis (summarised in refs. [1,20,21]).

Unlike those of dMOB1, the biological functions of dMOB2, dMOB3 and dMOB4 are yet to be completely understood. Based on the work by the Adler laboratory, it seems that dMOB2 can play a role in wing hair morphogenesis, possibly by forming a complex with Trc [4]. However, the precise mechanism of action remains unknown. Moreover, dMOB2 supports the development of photoreceptor cells [22] and the growth of larval neuromuscular junctions in flies [23]. It is possible that these neurological roles are linked to the association of dMOB2 with Tricornered [4]. So far, only Trc has been established as a bona fide binding partner of dMOB2.

Interestingly, like dMOB1 and dMOB2, dMOB3 can also genetically interact with Trc [4], but any function or direct binding partners of dMOB3 have remained elusive. Similar to dMOB2, dMOB4 has been linked to a neurological function. More precisely, dMOB4 was found to play a role as a regulator of neurite branching [24]. Furthermore, dMOB4 depletion in fly cells results in defective focusing of kinetochore fibres in mitosis [25]. Possibly some of these roles of dMOB4 could be linked to dMOB4 being part of the Striatin-interacting phosphatase and kinase (STRIPAK) complex (see subchapters 6 and 7 for more details). Yet, these possible connections have yet to be experimentally explored.

In summary, based on current evidence it is tempting to conclude that, unlike in yeast, individual dMOBs can interact (at least genetically) with Warts and Trc, the two NDR/LATS kinases expressed in fly cells. Conversely, individual NDR/LATS kinases can interact with different dMOBs. Therefore, it appears that in multicellular organisms such as flies, the binding of MOBs is not restricted to a unique member of the NDR/LATS kinase family.

## 3. An Overview of MOBs in Human Cells

hMOB1A and hMOB1B are also known as hMOB1, since hMOB1B is 96% identical to hMOB1A (see Figure 1 and refs. [1,3,26]). hMOB1A and hMOB1B are mainly cytoplasmic proteins [27,28]. However, upon targeting to the plasma membrane of mammalian cells hMOB1 can trigger a binding-dependent activation of the human NDR/LATS kinases [27,29]. hMOB1 can form complexes with NDR1 (aka STK38), NDR2 (aka STK38L), LATS1 and LATS2, all four human NDR/LATS kinases [1,13]. These interactions are mediated through one unique and highly conserved domain in NDR/LATS kinases [29]. Like dMOB1 [18], hMOB1 can also bind to the MST1/2 kinases [30], the human counterparts of Hpo [31,32,33,34]. The significance and regulation of these interactions is discussed in much detail in subchapters 5 and 6. hMOB1 can further associate with other proteins, although the importance of these additional interactions is yet to be defined (see subchapter 6). In cases whereby a new hMOB1 binding partner has been validated by conventional interaction assays, we have included the discussion of this novel aspect in the appropriate subchapters. Nevertheless, the protein-protein interactions of hMOB1 with NDR/LATS kinases is the best understood. For a summary of the cellular roles of hMOB1 please consult subchapter 7 and ref. [1].

In contrast to hMOB1, a significant portion of hMOB2 is nuclear [27,28]. Intriguingly, hMOB2 forms a complex with NDR1/2, while hMOB2 neither associates with LATS1/2 [1,35] nor MST1/2 [36]. hMOB2 can compete with hMOB1 for NDR1/2 binding, since hMOB2 interacts with the same domain on NDR1/2 as hMOB1 [27,28]. However, the binding modes of hMOB1 and hMOB2 to NDR1/2 seem to differ [1,35]. hMOB2 can also interact with RAD50, a core component of the MRE11-RAD50-NBS1 (MRN) DNA damage sensor complex, thereby playing a role in cell cycle-related DNA damage signalling [37]. Whether the hMOB2/RAD50 complex is linked to NDR1/2 signalling is yet to be defined [38]. For a summary of the cellular roles of hMOB2, see subchapter 7 and refs. [1,38].

Based on their similarities hMOB3A, hMOB3B and hMOB3C are sometimes also referred to as hMOB3 [39]. Like hMOB1, hMOB3s are mainly cytoplasmic proteins [28,40]. The best-understood binding partner of hMOB3s is the MST1 kinase [39]. Given the high sequence similarities between hMOB3s and hMOB1 (Figure 1), it was rather surprising that hMOB3s do not bind to any NDR/LATS kinase [35]. Nonetheless, large-scale interactome studies suggest that hMOB3s may have other binding partners in addition to MST1/2 (see ref. [1] and this review). Our current understanding of the cellular roles of hMOB3s is summarised in subchapter 7.

hMOB4 (aka Phocein) is best known as part of the multi-component STRIPAK (Striatin-interacting phosphatase and kinase) complex [41,42,43,44]. hMOB4 can bind to two distinct regions of Striatin [45]. Besides the Striatin-associated hMOB4, the core STRIPAK complex contains serine/threonine protein phosphatase 2 (PP2A) subunits such as PP2Ac and PP2A-A, as well as the Striatins as PP2A regulatory subunits. STRIPAK also encompasses CCM3 (cerebral cavernous malformation 3; also known as PDCD10) as well as the MST3 (aka STK24), MST4 (aka MASK or STK26) and STK25 (aka YSK1 or SOK1) protein kinases, the three members of the GCKIII (germinal centre kinase III) subfamily of Sterile 20 kinases [46]. In addition, the core complex contains STRIP1/2 (aka FAM40A/B). Two mutually exclusive STRIPAK complexes have been defined based on additional components such as CTTNBP2-like adaptors, SLMAP (sarcolemmal membrane-associated protein) or others [47]. However, although the STRIPAK complex has been linked to the Hippo pathway (see subchapter 4), the precise role(s) of hMOB4 in the STRIPAK complex has remained of a more speculative nature. In this regard, hMOB4 has been reported to form a complex with MST4 in a phosphorylation-dependent manner [48], an aspect that is discussed briefly in subchapter 6. The cellular functions of hMOB4 are discussed in subchapter 7.

## 4. An Overview of MOBs and the Hippo Pathway

The Hippo tissue growth control and regeneration pathway represents a key signalling cascade in flies and mammals [16,49,50,51]. By co-ordinating death, growth, proliferation, and differentiation on the cellular level the Hippo pathway controls organ growth, tissue homeostasis and regeneration. Hence, the deregulation of Hippo signalling has been linked to serious human diseases, such as cancers [50,52,53]. At the molecular level, a conserved Hippo core cassette is fundamental for the regulation of the co-transcriptional regulators YAP and TAZ, two main effectors of the Hippo pathway [51,54]. To date, the MST1/2 (aka STK4/3; Hpo in *D. melanogaster*) and LATS1/2 (aka Warts in flies) protein kinases are the best understood members of the Hippo core cassette. In this regard, MOB1 acts as central signal transducers in the Hippo core cassette. Upon activation of the Hippo pathway MST1/2 phosphorylate LATS1/2 and MOB1, thereby supporting the formation of an active MOB1/LATS complex (see subchapter 6 for molecular details), which is essential for development and tissue growth control [6]. Activated LATS1/2 in complex with MOB1 then phosphorylate YAP/TAZ on different Ser/Thr residues, thereby inhibiting nuclear activities of YAP/TAZ through cytoplasmic retention and/or degradation of phosphorylated YAP/TAZ [51,54]. Markedly, the NDR1/2 protein kinases (aka STK38 and STK38L; the closest relatives of LATS1/2 [13]), can also phosphorylate and thereby constrain YAP1 to the cytoplasm [55,56]. Similar to LATS1/2, NDR1/2 can stably associate with MOB1 [1,6] and are direct effector substrates of MST1/2 [55,57]. As already mentioned, MST1/2 also phosphorylate MOB1 [30]. Hence, LATS1/2, NDR1/2 and MOB1 represent well-established substrates of MST1/2 [1,55,57,58]. Taken together, the core of the Hippo pathway comprises distinct kinases, such as MST1/2, LATS1/2 and NDR1/2, which act together with the fundamental signal adaptor MOB1.

By directly interacting with the MST1/2, LATS1/2 and NDR1/2 kinases (aka Hpo, Warts and Trc in flies) MOB1 functions as a key signal transducer in the Hippo pathway. However, although dMOB1 and MOB1 are essential in flies and mice [6,7,59], hMOB1 appears to be dispensable in human cells, at least in transformed HEK293A cells [60]. hMOB1A/B double-knockout (DKO) cells were viable in spite of drastically impaired LATS1/2-mediated YAP/TAZ phosphorylation upon serum deprivation [60]. Very similar to LATS1/2 DKO cells, YAP/TAZ remained nuclear and co-transcriptionally active in hMOB1A/B DKO cells. Even more puzzling, hMOB1 phosphorylation by MST1/2 does not seem to be required for LATS1/2 activation in HEK293A cells [60], although biochemical evidence strongly suggests that MST1/2-mediated phosphorylation of hMOB1 is needed for LATS1/2 binding (see subchapters 5 and 6). To make things even more complicated, hMOB1 can associate with other intracellular regulatory proteins besides binding to kinases of the Hippo core cassette (see subchapter 6). Therefore, it is quite possible that hMOB1 as a central Hippo component is acting in diverse cancer-associated cellular processes.

Until recently [37], NDR1/2 were the only reported binding partners of hMOB2 [1,35]. More specifically, it was documented that hMOB2 could compete with hMOB1 for NDR1/2 binding, with hMOB2 counteracting hMOB1 as a co-activator of NDR1/2 [1,35,61]. In this context, Zhang et al. studied Hippo signalling in a hMOB2 knockout hepatocellular carcinoma (HCC) cell line [62]. They found that MST1/2-mediated phosphorylation of NDR1/2 was induced, while MST1/2-mediated phosphorylations of hMOB1 and LATS1/2 were decreased upon hMOB2 knockout [62]. LATS1/2-mediated phosphorylation of YAP was also reduced in hMOB2 knockout cells, while MOB2 overexpression resulted in opposite effects [62]. Subsequently, the authors concluded that loss of hMOB2 can favour hMOB1 binding to NDR1/2, thereby reducing the pool of hMOB1 available for LATS1/2 binding. Conversely, hMOB2 overexpression may reduce hMOB1 binding to NDR1/2, thereby freeing hMOB1 to bind to LATS, resulting in the activation of Hippo signalling upstream of YAP [62]. However, this study [62] did not examine whether this molecular reprogramming of binding partners could be a possible consequence of cell cycle effects upon hMOB2 loss-of-function. Given that loss of hMOB2 can trigger a p53-dependent G1/S cell cycle arrest [37] and that NDR/LATS are associated with cell cycle progression [20,21], it will certainly be necessary to re-examine these HCC-based hMOB2 knockout cells to ensure the changes in NDR1/2, LATS1/2, and hMOB1 phosphorylation upon hMOB2 loss [62] are not merely reflecting indirect effects triggered by an underlying cell cycle arrest/delay.

Like hMOB1 and hMOB2, the group of hMOB3 signal transducers has been linked to the Hippo pathway. It was found that hMOB3s can directly bind to MST1, while they do not bind to any member of the NDR/LATS kinase family [35,39]. Considering that hMOB3s share high sequence similarities with hMOB1 (Figure 1) this was unexpected. Nevertheless, hMOB3s have at least one aspect in common with hMOB1, namely the direct binding to the MST1/2 kinases. More specifically, hMOB3s require two conserved positively charged residues to bind to MST1 [39], with both positively charged residues also being essential for the formation of a stable complex between hMOB1 and MST1/2 [6]. However, current evidence suggests that hMOB3 binding to MST1 is inhibitory and hMOB3s can act upstream of apoptotic MST1 signalling [39], while hMOB1 seems to mainly function downstream of MST1/2 [1,30,51]. Therefore, it is possible that hMOB3 competes with hMOB1 for MST1/2 binding, like hMOB2 is competing with hMOB1 for NDR1/2 binding, but these competitive interactions are likely to involve different interaction domains (see subchapter 6).

Intriguingly, hMOB4 has likewise been connected with the Hippo pathway. In fly and human cells, the hMOB4-containing supramolecular STRIPAK complexes [43,63] have been implicated in Hippo signalling on diverse regulatory levels [48,64,65,66,67,68,69]. However, to our knowledge, the specific contributions of hMOB4 to the crosstalk between Hippo and STRIPAK signalling remain undetermined. Nevertheless, the MST4 kinase, a relative of MST1/2 [46], was shown to form a complex with hMOB4, with the overall structure of the hMOB4/MST4 complex resembling the hMOB1/MST1 complex [48]. But one of the key residues (Lys105) that promotes MOB1 binding to MST1 [6] is not conserved in hMOB4 (Figure 3). Notably, the phospho-Ser/Thr binding motif of hMOB1 [70] seems to be present in hMOB4, while key residues such as Glu51 of MOB1 that mediate NDR/LATS binding [6,71,72] are not conserved in hMOB4 (Figure 3). Therefore, the binding mode of MST4 and other GCKIIIs to hMOB4 is yet to be completely understood [73].

Different members of the Ste20-like kinase family can phosphorylate, and thereby activate, NDR/LATS kinases (summarised in ref. [57]). Briefly, MST1/2 phosphorylate NDR/LATS in human cells. Moreover, MAP4K (mitogen-activated protein kinase kinase kinase kinase)-type kinases can regulate NDR/LATS kinases through phosphorylation. In addition, MST3, MST4 and STK25, all members of the GCKIII subfamily of Ste20-like kinases [46], can phosphorylate NDR kinases in human and fly cells [57,74]. Strikingly, hMOB1 can bind to MST1/2, while hMOB4 is part of the GCKIII-containing STRIPAK complex. In this regard, the MAP4K-type kinase MINK1 was also identified as a component of the STRIPAK complex [75]. Consequently, it will be very interesting to properly map the binding patterns of hMOBs across the Ste20-like kinase family. Potentially, an array of different Ste20-like kinases is regulated by and/or regulates diverse hMOBs upstream of NDR/LATS signalling.

## 5. Post-Translational Modifications (PTMs) of MOBs

As already exemplified by the description of the Hippo pathway, signal transduction cascades transmit extra- and intra- cellular inputs through a broad array of protein kinases [76]. Thus, the phosphorylation status of signal transducers and effectors is finely orchestrated by interactions between kinases and phosphatases to achieve pathway regulation. Consequently, it is not surprising that MOBs can be regulated by phosphorylation events. In this subchapter, we summarise current knowledge on specific phosphorylation events of MOBs. Where appropriate, we also mention PTMs other than phosphorylations.

About two decades ago, yeast Mob1p was already described as a phospho-protein [5] that can be phosphorylated by Cdc15p (the yeast counterpart of the fly Hippo kinase) [77]. The phosphorylation of dMOB1 by Hippo was subsequently reported in fly cells [18]. However, Avruch and colleagues were the first to define the specific phosphorylation events that linked Hippo kinases such as MST1/2 with the regulation of hMOB1 [30,58]. More precisely, hMOB1 is specifically phosphorylated on Thr12 and Thr35 by MST1/2, thereby supporting complex formation between hMOB1 and NDR/LATS kinases [30]. Considering that the then available crystal structure of hMOB1A did not cover the N-terminal 32 residues [3], the precise molecular mechanism remained poorly understood for a long time [1]. However, Kim et al. recently deciphered the high-resolution crystal structure of full-length hMOB1 in its auto-inhibited form, and further reported the structural composition of a complex between hMOB1 and the N-terminal regulatory domain (NTR) of LATS1 [71]. These structures enabled the Hakoshima laboratory to define how hMOB1 is released from its auto-inhibitory conformation [71]. They found that hMOB1 phosphorylation on Thr12 and Thr35 accelerates the dissociation of the Switch helix, a long flexible positively-charged linker, from the LATS1-binding surface within hMOB1. Consequently, they uncovered a “pull-the-string” mechanism through which hMOB1 is rendered accessible for LATS1 binding [71], a model supported by an independent recent study [78]. This discovery was significant, since it helped to understand an important aspect of the molecular regulation of the core of the Hippo pathway [51,72].

Of note: in addition to the phosphorylation of Thr12 and Thr35 by MST1/2 [30], other PTMs of hMOB1, such as the phosphorylation of Tyr26 and Ser38, have been reported [79]. While the function of Ser38 phosphorylation has remained unclear, Tyr26 phosphorylation of hMOB1 has very recently been attributed an important role in tyrosine kinase signalling [80]. Gutkind and colleagues found that focal adhesion kinase (FAK) phosphorylates hMOB1 on Tyr26, thereby preventing the formation of a functional hMOB1/LATS complex and thus causing Hippo signalling to remain inactive [80]. It remains to be tested whether this new exciting regulatory mechanism of hMOB1 by Tyr26 phosphorylation also affects hMOB1/NDR complex formation. In addition, one should note that Ser23 of hMOB1 has also been observed [79], with Ser23 of hMOB1 representing a possible ATM target site [81], suggesting a possible link of hMOB1 to the DNA damage response (DDR). Last, but not least, it has been reported that GSK3β can destabilize hMOB1 by phosphorylating serine 146 of hMOB1 in the context of neurite outgrowth downstream of the PTEN-GSK3β axis [82]. Therefore, hMOB1 can be regulated by different phosphorylation events performed by diverse kinases.

In contrast to hMOB1, the only reported PTMs for hMOB2 are modifications with ubiquitin (or NEDD8) at Lys23, Lys32 and Lys131 [83,84]. The precise nature and functions of these three PTMs are currently unknown. In addition, the hMOB2 protein contains several putative ATM phosphorylation sites, which possibly are linked to the DDR function of MOB2 [81]. However, phosphorylation of hMOB2 has yet to be documented.

Similar to hMOB1, several phosphorylations of hMOB3s have been observed. Phosphorylations of hMOB3A on Thr15, Thr26 and Ser38, hMOB3B on Thr25 and Thr77, and hMOB3C on Thr14, Thr25 and Ser37 have been documented [79]. Ser37 of hMOB3C may represent an ATM targeting site [81]. Of note, the sequence motifs surrounding Thr15 and Ser38 of hMOB3A are quite similar to Thr12 and Thr35 of hMOB1 (Figure 3), hence raising the possibility that hMOB3A is possibly phosphorylated on Thr15 and/or Ser38 by MST1/2 in a similar fashion as reported for hMOB1 [30]. However, regulatory roles for any of these putative phosphorylation events are yet to be reported.

Last but not least, specific PTMs of hMOB4 are also of potential interest. In this regard, Ser147 and Tyr141 phosphorylations of hMOB4 have been observed [79]. Ser147 phosphorylation of hMOB4 is possibly performed by the DDR-linked ATM kinase [81], proposing that hMOB4 is linked to DNA damage signalling. However, no regulatory roles for any of these phosphorylation events have been described so far.

## 6. Protein-Protein Interactions of MOBs

Given that MOBs do not have any known enzymatic activities, one wonders how they contribute to intracellular signalling. Based on the currently available knowledge, the answer to this question is that MOBs are globular scaffold proteins that can regulate other factors such as protein kinases by direct protein-protein interactions [1]. Therefore, we will focus in this subchapter on summarising how MOBs form protein complexes and thereby help to regulate molecular machineries.

As already mentioned, hMOB1 can bind directly to MST1/2, LATS1/2 and NDR1/2. Interestingly, these interactions can occur in a phosphorylation-dependent manner. However, before discussing this important aspect in much detail, one should first note that hMOB1 can also associate with other signalling complexes. Current evidence suggests that hMOB1 can bind to at least seven apparently mutually exclusive and independent proteins in a sometimes phosphorylation-dependent fashion: (1) the NDR1/2 kinases, (2) the LATS1/2 kinases, (3) the MST1/2 kinases, (4) the PP6 phosphatase module, (5) the atypical guanine nucleotide exchange factors 6 to 8 (DOCK6-8), (6) the ubiquitin ligase Praja2, and (7) yet poorly characterized proteins, including leucine rich repeats and calponin homology domain containing (LRCH1-4) proteins and cytokine receptor like factor 3 (CRLF3) [1,6,30,65,78,85,86]. Notably, Xiong et al. found that the phosphorylation-dependent binding of hMOB1 to the PP6 and DOCK6-8 complexes appears to differ from that elucidated for protein kinase binding, with hMOB1 preferentially interacting with MST1/2 and NDR/LATS, and not the DOCK6-8 and PP6 modules [78]. Future investigations are needed to properly understand the underlying cellular functions of complexes containing hMOB1 and non-kinase binding partners such as PP6, DOCK6-8 and others.

While hMOB1 can bind to a variety of components, the complex formations with MST1/2 and NDR/LATS kinases are the only protein-protein interactions understood in detail. In Hippo core signalling, hMOB1 acts as a central signal adaptor helping to regulate NDR/LATS activation [1,6,13,71,72,87,88]. On the one hand, a stretch composed of highly conserved hydrophobic and positively charged residues N-terminal of the catalytic domains of NDR/LATS kinases, termed N-terminal regulatory domain (NTR), is essential for hMOB1 binding to NDR/LATS kinases to promote their activities. On the other hand, a conserved cluster of negatively charged residues on hMOB1 is needed for the interaction with NDR/LATS [1,3,6,13,28,71,72,87,88]. MST1/2 phosphorylation of hMOB1 on Thr12 and Thr35 regulates hMOB1 binding to the NTRs of NDR/LATS [6,30,71,72], although hMOB1 can also from a complex with NDR1/2 in a phosphorylation-independent manner [6]. In summary, hMOB1 binds to NDR/LATS kinases through highly conserved residues on hMOB1 and NDR/LATS kinases in a manner that can be regulated by MST1/2 phosphorylation of hMOB1.

hMOB1 binding to NDR/LATS has multiple functions. First, hMOB1 binding helps to stimulate auto-phosphorylation on a Ser residue located in the T-loop (also known as activation segment) of NDR/LATS kinases (summarised in [1,13,87,89]). Second, MOB1 also acts as an allosteric activator of NDR/LATS by mediating MST1/2-mediated trans-phosphorylation of a Thr residue located in the hydrophobic motif (HM) outside of the catalytic domain of NDR/LATS [6,28,78,87,88,89,90,91]. Third, it was biochemically shown that stable hMOB1 binding to MST2 can function as an important step in activating the MST1/2-LATS1/2 kinase cascade [72,92], although stable MOB1/MST1/2 binding seems dispensable for development and tissue growth control [6,90] and a stable hMOB1/MST1/2 complex is not required for efficient MST1/2-mediated phosphorylation of hMOB1 [6]. Since some of these hMOB1 functions appear to contradict each other, Manning and Harvey [93] proposed a unifying model, wherein hMOB1 acts before and after MST1/2-mediated phosphorylation of LATS1/2 and hMOB1. Notably, this model is similar, but still different, from the model proposed for NDR1/2 activation by MST1/2 and hMOB1 [6,88]. In this regard, somewhat surprisingly, T-loop and HM phosphorylations of LATS1/2 can also occur without increased MST1/2-mediated phosphorylation of hMOB1 on Thr12/Thr35 [94], suggesting that hMOB1/LATS complex formation can be dispensable for efficient LATS1/2 phosphorylation in certain settings. Nevertheless, hMOB1/LATS complex formation appears to be essential for development and tissue growth control [6].

Taken together, these data show collectively that hMOB1/NDR and hMOB1/LATS complexes are not equal, thereby allowing cells to process diverse biological inputs through diverse hMOB1 signalling. hMOB1 as an adapter protein can help to differentially regulate NDR/LATS kinases by concurrently forming complexes with NDR/LATS and their upstream activator kinases such as MST1/2 and by allosterically activating the auto- and trans-phosphorylation activities of NDR/LATS.

Interestingly, recent structural and biochemical analyses of the catalytic domain of NDR1, lacking the entire NTR domain, have suggested that the regulation of NDR1 activity by the elongated activation segment and by hMOB1 binding may represent mechanistically distinct events [89]. This notion is supported by the biochemical and structural dissection of the yeast Cbk1p-Mob2p complex [95]. However, much more work is needed to fully understand how the regulation of NDR/LATS kinases is orchestrated by hMOB1 binding to the NTR, activation segment and HM phosphorylations–three molecular events that possibly represent three distinct regulatory levels. In particular, the role of MST1/2-mediated phosphorylation of hMOB1 is yet to be completely understood with regard to the regulation of NDR1/2, in particular when considering that phospho-hMOB1 displays a much higher affinity for NDR1/2 than unphosphorylated hMOB1 [6,78], while LATS1/2 can only bind to phospho-MOB1 [6].

In contrast to NDR1/2 kinases, hMOB1 seems to associate with LATS1/2 and MST1/2 in a purely phospho-dependent manner [6,65,69,70,71,72,78,92]. In the case of MST1/2, the direct interactions involve several auto-phosphorylation sites in MST1/2 that help to recruit hMOB1 [51,65,69,72,78,92]. Noteworthy, these interactions involve a phospho-Ser/Thr binding pocket of hMOB1, encompassing the positively charged residues Lys153, Arg154 and Arg157 of hMOB1 [70]. These three key residues appear to be conserved in hMOB1 and hMOB4, while hMOB3s have only two conserved positively charged residues (Figure 3). However, phospho-peptide binding properties of hMOB2, hMOB3s and hMOB4 are yet to be reported.

Based on the structure of a short fragment of MST2 (371 to 400) bound to hMOB1 lacking the first 50 amino acids, Ni et al. [72] defined two MOB1/MST2 interfaces, namely one supportive binding site (residues 390 to 398 of MST2) and one main phospho-Thr binding site, centring around phosphorylated Thr378 of MST2 and the phospho-Ser/Thr binding pocket of hMOB1. However, other important sites involved in hMOB1 binding on MST2, such as phospho-Thr349, phospho-Thr356, and phospho-Thr364, were not examined on the structural level [72]. This notion [72] that Thr378 of MST2 (Thr380 of MST1, respectively) is not the sole main interaction site on hMOB1 is supported by other studies [78,92]. At least six different phospho-threonine sites on MST1 (Thr329, Thr340, Thr353, Thr367, Thr380 and Thr387) contribute to hMOB1 binding [78,92]. Intriguingly, Pro106 of hMOB1 contributes to phospho-Thr binding on MST1 [72,78,92], and the positively charged Lys104 and Lys105 of hMOB1 are crucial for MST1/2 binding, while being expendable for NDR/LATS binding [6]. Thus, it is quite possible that Lys104/Lys105 together with Pro106 of hMOB1 may bond with multiple negatively charged phosphorylated Thr residues on MST1/2. Notably, the Lys104/Lys105/Pro106 motif of hMOB1 is conserved in dMOB1, hMOB3s and dMOB3 (Figure 3). Alterations of Arg108 and Lys109 in hMOB3A, corresponding to Lys104 and Lys105 in hMOB1 (Figure 3), abolish MST1 binding [39], suggesting that the Lys104/Lys105/Pro106 binding motif of hMOB1 is functionally conserved.

Notably, phospho-Thr378 of MST2 can also bind to SLMAP [68], in addition to hMOB1 [69], and a T-loop mutation of MST2 reduces the MST2/SLMAP interaction [65] in the context of STRIPAK-Hippo signalling. Thus, the multisite auto-phosphorylation of the MST1/2 linkers appears to function as a molecular platform able to integrate different Hippo signalling events [68,69].

In contrast to MST1/2, the LATS1/2 kinases must not be phosphorylated to bind to phospho-MOB1 [6,71,72]. Nevertheless, the phosphorylation of LATS1/2 on serine 690 and 653, respectively (corresponding to threonine 74 and 75 in NDR1 and NDR2), are very likely to represent another regulatory layer. LATS1 S690A, LATS2 S653A, NDR1 T74A and NDR2 T75A mutants display diminished binding to MOB1 [27,28,29,87,88,96,97] and the crystal structure of phospho-MOB1 bound to a LATS1 fragment highlighted the importance of serine 690 of LATS1 for MOB1 binding [72]. In this regard, hMOB1 must be phosphorylated to bind to LATS1/2 [6,71,72], with Asp63 of hMOB1 being essential and providing the selectivity for LATS1/2 binding [6]. In this regard, the analysis of hMOB1 mutants with alterations of Lys153 and Arg154 revealed that the phospho-Ser/Thr binding interface of hMOB1 is important for MST1/2 as well as LATS1/2 binding [6]. Consequently, Lys153 and Arg154 of hMOB1 are very likely to represent central residues in a more general phospho-Ser/Thr binding domain as already proposed by Rock et al. [70]. Notably, Lys153 and Arg154 of hMOB1 can also contribute to Praja2 binding in the context of hMOB1 proteolysis [86].

In addition, the interactions of hMOB1 with NDR/LATS kinases can be modulated by Tyr26 phosphorylation of hMOB1 [80] and hMOB2 counteracting hMOB1 binding to NDR1/2 [35,62]. Comparable to hMOB2, the LIM domain protein TRIP6 can compete with MOB1 for binding to the NTR of LATS1/2, thereby negatively interfering with LATS1/2 activation by hMOB1 [98]. Similarly, phosphatidic acid-related lipid signalling can regulate the Hippo pathway by directly interacting with LATS1/2, thus disrupting hMOB1/LATS complex formation [99]. In this context, one should note that inactivation of the PTEN lipid phosphatase can interfere with the interaction of hMOB1 and LATS1/2 in gastric cancer [100]. Moreover, the Angiomotin scaffolds can support hMOB1-mediated LATS1/2 auto-phosphorylation [101].

The best understood binding partners of hMOB2, hMOB3s and hMOB4 were already mentioned in subchapters 3 and 4. Briefly, two binding partners of hMOB2 have been established: the NDR1/2 kinases [1,35] and the RAD50 scaffold [37]. However, the physiological significance of these interactions is yet to be fully defined [38], with hMOB2 possibly also having indirect effects on LATS1/2 signalling by competing with hMOB1 binding to LATS1/2 [35,62].

hMOB3s have one established binding partner: the MST1 kinase [36]. In response to apoptotic stimuli, hMOB3s bind to MST1 through conserved positively charged residues, thereby negatively regulating apoptotic MST1 signalling in glioblastoma multiforme cells by inhibiting the MST1 cleavage-based activation process [36]. In this regard, it is crucial to note that hMOB3s do not function like hMOB1 in the Hippo pathway as proposed recently [102]. While hMOB3s and hMOB1 share similar primary sequences (Figure 1 and Figure 3), they differ significantly in their binding partners, and hence their potential to regulate the core of the Hippo pathway. Nevertheless, one cannot rule out the possibility that hMOB3 and hMOB1 may compete for MST1/2 binding, similar as already reported for the competition of hMOB2 and hMOB1 for NDR1/2 binding [1,35].

As mentioned, hMOB4 is a core member of the STRIPAK complex, known to function as a regulator of Hippo signalling [43,63]. Recently, it was reported that hMOB4 forms a complex with MST4 [48], a distant relative of MST1/2 [46] and a known component of the STRIPAK complex [73]. It was proposed that MST4 and hMOB4 may disrupt assembly of the hMOB1/MST1 complex through alternative pairing [48]; however, this model has yet to be explored in more detail.

Taken together, we have started to understand the mechanistic importance of protein-protein interactions involving hMOBs. However, some of our current working models have yet to be consolidated by future studies, ranging from more structure biology to the testing of the significance of selected protein-protein interactions in adequate biological model systems. In particular, we need to comprehend how phosphorylation-triggered interactions of hMOBs help to coordinate the specificity and robustness of essential signalling cascades. In this context, it will be vital to understand the potential functional contribution of the abilities of hMOBs to engage with multiple ligands.

## 7. Cancer-Associated Cellular Functions of MOBs

### 7.1. Roles in Mitosis and Cell Cycle Progression

Since we are only briefly summarising here the mitotic functions of hMOB1, for a detailed overview of the cell cycle roles of hMOB1 see refs. [20,21]. Initially, a kinase-targeting RNAi screen found that the NDR1 kinase, a binding partner of hMOB1 and hMOB2 [1], is possibly involved in regulating spindle orientation [103]. Yao and colleagues subsequently discovered a new level of hMOB-mediated NDR1 regulation in mitosis [61]. More precisely, PLK1 phosphorylates NDR1 at Thr7, Thr183 and Thr407 upon mitotic entry, thereby eliciting PLK1-dependent suppression of NDR1 activity to ensure correct spindle orientation in mitosis [61]. Mechanistically, PLK1-mediated phosphorylation of NDR1 switches NDR1 binding from hMOB1 to hMOB2, thereby interfering with the binding of hMOB1 to NDR1 and subsequent NDR1 activation [61]. The involvement of hMOB1 in mitosis is further supported by the finding that hMOB1 knockdown displays a negative genetic interaction with the *PTTG1* gene [104], with *PTTG1* encoding Securin, a well-established regulator of chromosome segregation [105]. Noteworthy, hMOB1 can localise to kinetochore structures of mitotic cells [106] and contributes to cell abscission and centriole re-joining after telophase and cytokinesis [107].

Knockdown of hMOB4 can also cause mitotic spindle defects in human cells [108]. Similarly, dMOB4 depletion results in mitotic defects in fly cells [25]. However, it is currently not known whether this function is related to the MOB4-containing STRIPAK complex. In contrast to hMOB1 and hMOB4, the depletion of hMOB2 results in a G1/S cell cycle arrest [37,109]. More specifically, hMOB2 knockdown triggers a p53-dependent G1/S cell cycle arrest [37] that is discussed in more detail in subchapter 7.2. hMOB3s have no known role in mitosis.

### 7.2. Roles in the DNA Damage Response (DDR)

Considering the possible involvement of NDR1/2 in the DDR [38], one would predict that regulators of NDR1/2, such as hMOB1 [1], may also play a role in the DDR. Interestingly, this seems to be the case, since hMOB1 knockdown appears to be sufficient to cause spontaneous DNA double-strand break formation [110] and increased γH2AX phosphorylation in human cells [111]. The single knockdown of hMOB3A or hMOB4 can also result in increased γH2AX phosphorylation in cells [111], while hMOB3C knockdown cells seem to display an increased homologous recombination repair frequency [112]. Collectively, these reports propose that hMOB1, hMOB3 and hMOB4 are possibly linked to DNA damage signalling. However, it is yet to be established whether any of these possible DDR functions are linked to binding to kinases of the Hippo core, such as NDR/LATS, or components of the STRIPAK complex.

Regarding hMOB2 we have a somewhat clearer understanding regarding the DDR [38]. Given that a genome wide screen for novel putative DDR factors identified hMOB2 (aka HCCA2) as a potential candidate [113], we sought to understand how hMOB2 might function as part of the DDR. Subsequent experiments uncovered that loss of hMOB2 causes the accumulation of DNA damage, which activates the DDR kinases ATM and CHK2, consequently activating a p53 dependent G1/S cell cycle checkpoint in the absence of any exogenously induced DNA damage [37]. Complementary experiments showed that hMOB2 promotes cell survival and G1/S cell cycle arrest upon exposure to diverse DNA damaging agents [37]. Unexpectedly, hMOB2 seems to act in the DDR independent of NDR signalling [37], therefore, the biological significance of the hMOB2/NDR complex remains poorly understood [38]. Mechanistically, hMOB2 can interact with RAD50, facilitating the recruitment of the MRE11-RAD50-NBS1 (MRN) DNA damage sensor complex and activated ATM to DNA damaged chromatin [37]. However, this mechanistic link does not appear to be relevant in all DDR settings, suggesting that additional mechanisms should be considered [38].

Of note, other independent studies support the notion that hMOB2 represents a new player in the DDR. hMOB2 knockdown cells displayed an increased sensitivity to exogenously induced DNA-damage and a defective G2/M cell cycle checkpoint in response to ionizing radiation (IR) [113]. Furthermore, patient-derived fibroblasts with MOB2 loss-of-function exhibited increased susceptibility to exogenously induced DNA-damage [114]. hMOB2 knockdown in non-small cell lung cancer (NSCLC) cell lines is also a promising strategy to increase the susceptibility to IR in the context of radio-sensitization [115]. In addition, the search for mutator alleles which increase the rate of germline mutations has revealed the *hMob2* gene as one of the top hits [116]. Thus, current evidence collectively suggests that the involvement of MOB2 in the response to DNA damage is important to maintain genomic stability. hMOB2 can be part of the DDR by playing roles in DDR kinase signalling, cell survival and cell cycle checkpoints upon exposure to DNA damage. However, we have yet to understand how hMOB2 operates as a DDR protein on the molecular level [38]. Furthermore, we must comprehend which types of DNA damage repair mechanism(s) are linked to hMOB2 and whether this knowledge can be clinically exploited for precision medicine.

### 7.3. Apoptosis and Autophagy

hMOB1 has been linked to apoptotic signalling [1]. hMOB1 as a co-activator of human NDR kinases plays an important role in the formation of a MST1-NDR-hMOB1 complex, thereby supporting MST1-mediated phosphorylation of NDR1/2 upon the induction of apoptosis [91,117]. Interestingly, the apoptotic role of hMOB1 was further linked to cell fate decisions in response to autophagic stress conditions. Precisely, upon investigating the activation of NDR1 in macroautophagy, we observed that NDR1 is stimulated in a hMOB1-dependent manner upon autophagy induction [118]. Considering further that NDR1′s kinase activity also plays an important role in mitophagy, the removal of damaged mitochondria by selective autophagy [119], it is likely that hMOB1 as a co-activator of NDR1/2 possibly functions in different autophagic processes. Taken together, hMOB1 seems to assist NDR1/2 to coordinate autophagic and apoptotic events.

### 7.4. Centrosome Biology

By functioning as a co-activator of human NDR kinases, hMOB1 can also play a role in centrosome duplication [28]. However, although hMOB1 can localise to centrosomes [106,120], we do not know whether the centrosomal pool of hMOB1 is the main supporter of centrosome duplication. Based on data from a large-scale screen hMOB1 appears to bind to DIPA [121,122], which is of potential interest, considering that DIPA was found on centrosomes [123]. Moreover, the NDR2 kinase was recently connected with ciliogenesis, a cellular process that utilises the mature mother centriole at the base of cilia extrusions [124,125]. Considering further that active (phosphorylated) NDR1/2 can localise to the mother centriole in a cell cycle dependent fashion [28], it is quite tempting to speculate that hMOB1 as a main regulator of NDR1/2 is linked to the ciliogenesis. This notion is supported by additional lines of evidence:

First, upon studying different canine models for retinal degeneration, it was found that hMOB1 expression was specifically elevated in the early retinal degeneration model [126], an autosomal recessive disorder caused by a NDR2 mutation [127]. Possibly this reflects the cellular attempt to compensate for NDR2 loss-of-function. Second, murine NDR1/2 kinases are important for regulation of proliferation of terminally differentiated cells in the retina [128]. Third, increased levels of phospho-Thr35 MOB1 were observed in a fish model for photoreceptor ciliogenesis and retinal development [129]. Fourth, MOB1 accumulates at the basal bodies and MOB1 depletion delays ciliogenesis in *Tetrahymena* [130]. Therefore, MOB1 appears to represent a Hippo component that provides a link between the Hippo pathway and ciliogenesis in the context of diseases such as retinal degeneration and possibly also polycystic kidney disease [131,132,133,134,135,136,137,138,139].

### 7.5. MOBs and RAS Onco-Proteins

Oncogenic Ras signalling occurs frequently in many human cancers. However, no effective targeted therapies are currently available to treat patients suffering from Ras-driven tumours. In this regard, a systematic screen uncovered YAP1, a key effector of Hippo signalling [17,140], as a significant component of oncogenic RAS signalling [141]. Specifically, YAP1 overexpression is sufficient to rescue cell viability in a KRAS-dependent fashion [141]. Interestingly, the same screen also identified hMOB2 and hMOB3C as possible factors in oncogenic RAS signalling [141]. hMOB2 overexpression could decrease cell viability in a KRAS-dependent manner, while hMOB3C overexpression restored cell viability without any obvious effects on MEK and PI3K signalling [141]. Another genome-wide RNAi screen identified a synthetic lethal interaction between hMOB3A loss-of-function and oncogenic KRAS [142]. Perhaps the RAS-related function of hMOB2 is linked to hMOB2/NDR complex formation, since NDR1/2 were recently linked to oncogenic RAS signalling [119,143]. Future research is warranted to probe these interesting links.

## 8. MOBs and Cancer

### 8.1. Lung cancer

hMOB1A mRNA levels are low in human lung cancer samples [144]. Consequently, Suzuki and colleagues studied inducible conditional MOB1 knockout mice in the context of lung physiology and cancer, revealing that mice with postnatally MOB1 loss-of-function did not develop spontaneous lung adenocarcinomas [145]. Unexpectedly, urethane treatment-induced lung tumour formation was decreased upon postnatal MOB1 deletion [145], suggesting that MOB1 rather functions as a cancer-promoting factor in this setting. Thus, it will be very interesting to elucidate which functional protein-protein interaction(s) involving MOB1 perform this pro-cancer role. Noteworthy, hMOB3B seems to interact with NT5C2 [121]. Given that NT5C2 has been used as a prognostic marker in lung cancer [146], it is tempting to speculate that hMOB3s might be of clinical relevance in lung cancer.

### 8.2. Pancreatic Cancer

Chen et al. found that miR-181c can directly and selectively repress hMOB1, MST1, LATS2, and SAV1 expression in human pancreatic cancer cells [147]. Inhibition of miR-181c increased the protein levels of hMOB1 and other Hippo components. In contrast, overexpression of miR-181c suppressed them, resulting in increased pancreatic cancer cell survival and chemoresistance [147]. Moreover, deregulated expression of the long non-coding RNA (lncRNA) UCA1 has been implicated in diverse human cancers, such as pancreatic cancer. UCA1 can interact with hMOB1, LATS1, and YAP to form shielding composites, thereby allowing YAP activation in pancreatic cancer cells [148]. Thus, deregulation of hMOB1 as part of the Hippo pathway can play a significant role in the progression of pancreatic cancer. Of note, expression levels of hMOB4 were found elevated in pancreatic cancer samples [48]. Thus, diverse hMOBs are of potential interest in clinical studies of pancreatic cancer.

### 8.3. Liver Cancer

The Thr12 phosphorylation levels of hMOB1 were significantly decreased in human liver cancer samples [94]. Mice with liver-specific *Mob1* null alleles are prone to develop liver cancers [149]. Given that this phenotype is the most severe among mutant mice lacking a Hippo signalling component in the liver [150], it is fair to conclude that MOB1 constitutes a critical hub of Hippo signalling in livers and possibly also other tissues. In this regard, it is noteworthy that increased expression of hMOB2 mRNA was associated with hepatocellular carcinoma development and progression [151]. However, due to the lack of MOB2 loss-of-function animal models it is currently unclear whether deregulated MOB2 expression can drive liver cancer development.

Given that co-infections with hepatitis delta virus (HDV) and hepatitis B virus (HBV) accelerate the development of hepatocellular carcinoma (HCC) [152], one should also note that hMOB1 can associate with CCDC85B (aka DIPA) [121,122], which is known to play a role in HDV replication [153]. hMOB1 also interacts with the hepatitis C virus (HCV) non-structural protein NS5A, although no relationship between the hMOB1A-NS5A interaction and HCV replication has been observed [154]. Nevertheless, the hMOB1-binding partners LATS1/2 represent host kinases that can phosphorylate NS5A at a highly conserved residue required for optimal HCV replication [155].

### 8.4. Haematological Malignancies

By studying the effects of 3-deazaneplanocin A (DZNep), a histone methyltransferase EZH2 inhibitor, on human T-cell acute lymphoid leukaemia (T-ALL) cells, Shen et al. found that DZNep treatment reduced histone methylation in hMOB1 promoters, resulting in an upregulation of hMOB1 expression associated with inhibited growth of T-ALL cells [156]. Thus, MOB1 levels may represent an indicative marker for DZNep treatments of T-ALL patients.

As mentioned, hMOB3 may interact with NT5C2 [121]. Given that NT5C2 expression levels can serve as a prognostic marker in haematological malignancies [157,158] and NT5C2 mutations were linked to ALL [159], it is therefore tempting to deduce that a link between hMOB3s and NT5C2 might be of clinical significance with regard to haematological cancers. In this regard, one should further note that altered expression levels of hMOB3A and hMOB3B were observed in mantle cell lymphoma [160,161]. However, animal studies that functionally link the deregulation of MOB3 with haematological malignancies are missing to date.

### 8.5. Breast and Ovarian Cancers

Kim et al. found that hMOB1 knockdown decreased the viability of breast cancer cell lines [162], suggesting that hMOB1 is supporting the survival of breast cancer cells. Changes in the methylation of the hMOB1 promoter were observed in triple negative breast cancer cells [163]. Likewise, a systematic study across cancer cell lines revealed that MOB1 knockdown results in decreased viability of ovarian cancer cells [164], suggesting that hMOB1 is supporting the survival of ovarian cancer cells. Of note, single knockdown of hMOB3A, hMOB3B or hMOB3C also decreased the viability of breast cancer cells [162].

### 8.6. Colon Cancer

The mRNA levels of hMOB1A are low in human colorectal cancer samples [165]. Recently, a study of mice with intestinal epithelial cell-specific depletion of MOB1 uncovered a link between MOB1 and intestinal homeostasis through the regulation of Wnt signalling [166]. Considering that the disruption of intestinal homeostasis by the deregulation of other Hippo components can result in colon cancer [167], it is quite likely that loss of MOB1 as a central component of the Hippo pathway is also associated with colon cancer development and/or progression. Of note, single knockdown of hMOB3A, hMOB3B or hMOB3C in human colon cancer cells displays a negative genetic interaction with the PTEN phosphatase [104].

### 8.7. Prostate Cancer

A single nucleotide polymorphisms (SNP) in the *hMob2* gene has been revealed to be a prostate cancer susceptibility loci [168]. The mRNA levels of hMOB3B were significantly decreased in prostate cancer, suggesting that hMOB3B may act as a tumour suppressor in prostate cancer [169]. Moreover, the hMOB3B promoter region is also hypermethylated in a significant number of prostate cancer samples [170]. Thus, loss of hMOB3B might promote prostate cancer development and/or progression.

### 8.8. Glioblastoma

On the one hand, it was found that decreased hMOB1 levels help to sustain glioblastoma growth [86]. On the other hand, the protein levels of hMOB3s are significantly upregulated in samples derived from patients suffering from glioblastoma multiforme [36]. Therefore, it appears that hMOB1 and hMOB3s may be considered in clinical assessments of glioblastoma samples.

### 8.9. Human Cancer Cell Lines

Based on genome-scale loss-of-functions screens using a broad spectrum of diverse human cancer cell lines, one is tempted to conclude that the single knockdown or knockout of hMOB1A, hMOB1B, hMOB3A, hMOB3B, hMOB3C or hMOB4 can decrease the survival of some cancer cells [171,172,173,174]. However, these findings have yet to be validated, and even more importantly, are most likely limited in their nature, since possible compensatory redundancies in the MOB1 and MOB3 groups were not examined. Of note, the status of hMOB3B together with a few other markers can serve as predictor of acquired and intrinsic resistance to epidermal growth factor receptor tyrosine kinase inhibitors, including gefitinib, erlotinib, and afatinib [175].

## 9. MOBs and Other Diseases

### 9.1. Neurobiology

MOB2 loss-of-function impairs the correct neuronal positioning within the developing cortex, thereby associating MOB2 inactivation with a disorder such as periventricular nodular heterotopia [114]. Moreover, MOB2 plays a part in PKA-mediated signalling in astrocytes [176]. Similarly, MOB2 plays a neurological function in invertebrates, since dMOB2 expression is necessary and sufficient to regulate the growth of larval neuromuscular junctions in flies [23].

Chromatin immunoprecipitation followed by deep sequencing identified the hMOB3A gene as a target for the NRF1 (Nuclear respiratory factor 1), a transcription factor linked to neurodegenerative diseases [177]. Recombinant hMOB3A and hMOB3B can bind to β-amyloid, suggesting that hMOB3s are possibly linked to Alzheimer disease and other atrophies [178]. Interestingly, the same study found that hMOB1 can also associate with β-amyloid [178].

hMOB4 is mainly expressed in the central and peripheral nervous system [179]. Striatin-1, a core component of the STRIPAK complex, appears to require hMOB4 binding to fulfil its neurological function [180]. This notion is supported by the observation that dMOB4 also plays a role in the regulation of axonal transport, neurite elongation, and synapse formation in flies [24].

Taken together, each member of the MOB family seems to have at least one neurological function, with the neurological roles of MOB2 and MOB4 being the best understood.

### 9.2. Immunity and Virology

Quite likely hMOB1, based on its associations with NDR/LATS and MST1/2 kinases [1] is linked to immunological functions as summarised in ref. [57]. In addition, it was reported that the conserved *Legionella pneumophila* effector kinase LegK7 can phosphorylate MOB1 and thereby mimic MST1/2 activation to shut down the Hippo effectors YAP/TAZ, resulting in the promotion of bacterial growth and infection [181]. In this regard, the other hMOBs are possibly also associated with immune signalling in health and disease. More precisely, a SNP in the *hMOB2* gene is associated with altered cytokine levels, in particular interleukin 9 (IL-9) levels [182]. SNPs in the *hMOB3B* gene are associated with rheumatoid arthritis [183], osteoarthritis [184] and asthma [185]. Moreover, a genome-wide RNAi screen identified hMOB4 as a possible regulator of cytokine secretion, since IL-8 levels were decreased upon hMOB4 knockdown [186].

### 9.3. Diverse Disease Spectrum

In addition to links to neurological and immune diseases, the deregulation of hMOBs has also been associated with other sicknesses as briefly summarised below. MOB1A/B-deficient mouse epidermis cannot be engrafted successfully onto donor mice, thereby influencing skin engraftment efficiency [187], suggesting a crucial role of MOB1 in the skin. Mice with chondrocyte-specific MOB1A/B-deficiency display impaired chondrocyte homeostasis, leading to chondrodysplasia, thereby linking MOB1 loss-of-function with a hereditary skeletal disorder [188]. Furthermore, an autosomal SNP of *hMob2* is associated with type 2 diabetes [189,190]. Last but not least, it is notable that the dysregulation of the hMOB4-containing STRIPAK complex correlates with a broad range of human diseases including, diabetes, autism, cardiac disease and cerebral cavernous malformation (CCM) in addition to cancer [43,44]. However, it remains to be determined the degree to which hMOB4 is essential to support fundamental cellular processes such as cell proliferation, growth, differentiation, migration and apoptosis.

## 10. Conclusions and Future Outlook

In summary, yeast cells express two different MOBs that act in independent and non-interchangeable complexes formed with members of the conserved NDR/LATS kinase family. In contrast, in multicellular organisms such as flies and humans, MOBs do not exclusively bind to a unique NDR/LATS kinase. Fly cells express at least four different dMOBs, while the human genome encodes at least seven different *hMOB* genes. hMOBs have been implicated in different crucial intracellular functions through forming complexes with diverse binding partners (Figure 4; see also subchapter 6). By associating with members of the NDR/LATS kinase family, hMOB1 has central roles in Hippo signalling (Figure 4, left panel). Importantly, hMOB1 also binds directly to MST1/2 as part of the Hippo core cassette (Figure 4, left panel). By forming a complex with the MRN DNA damage sensor complex, hMOB2 has been linked to DNA damage signalling (Figure 4, middle left panel). Similar to hMOB1, hMOB3s can associate with MST1 in the context of apoptotic signalling, but unlike hMOB1, hMOB3s do not interact with NDR/LATS kinases (Figure 4, middle right panel). Last but not least, hMOB4 has been documented as a core member of the STRIPAK complex (Figure 4, right panel).

Although the four branches of MOB signalling can be depicted as four separate signal transduction cascades, one should note that it is quite likely that they are interconnected (Figure 4, see dashed grey lines). For example, it is known that hMOB2 can oppose certain functions of the hMOB1/NDR complex by competing with hMOB1 for NDR1/2 binding. Hence, it could well be that hMOB2 is interlinked with the Hippo pathway through interfering with hMOB1-regulated processes. Another example is the possible connection of hMOB3s with the Hippo pathway. Perhaps hMOB3s can compete with hMOB1 for MST1/2 binding and thereby meddle with hMOB1-regulated Hippo signalling. Furthermore, hMOB4 as part of the STRIPAK complexes may help to connect the GCKIII upstream kinases with NDR1/2 as members of the Hippo pathway. As is apparent from these examples, it is conceivable that hMOB2, hMOB3s and hMOB4 are interconnected with hMOB1-regulated Hippo signalling to some degree (Figure 4, see dashed grey lines). Certainly, future research is warranted to address these predicted crosstalks between different hMOB-containing complexes.

Some characteristics of MOB proteins are conserved from yeast to man, but MOB signalling is surely more complex in multicellular organisms. Therefore, one future challenge is to fully comprehend and appreciate this complexity. Despite great progress in understanding how MOBs can help to regulate their binding partners, much more remains to be accomplished in this regard. For instance, more research is needed to fully understand how hMOBs are fine-tuning the activities of NDR/LATS kinases, with particular emphasis on delineating the in vivo implications of MST1/2-mediated phosphorylation of hMOB1. In this context, we also must understand which protein-protein interactions of hMOBs are essential on the cellular and organismal levels. For example, multiple binding partners of hMOB1 have been reported, but we have only begun to appreciate the physiological importance of a fraction of these hMOB1-focused complexes. The molecular and biological understanding of the significance of hMOB2/NDR, hMOB2/MRN, hMOB3/MST1 and hMOB4/STRIPAK complexes has also remained a pressing question that is yet to be dissected in much detail. Possibly, forthcoming studies on MOBs from organisms such as fungi, parasites, bacteria and plants [1,181,191,192] will help to provide breakthroughs in this regard; studies of yeast MOBs have initially led the way [1,5,8].

Taken together, future research is needed to improve the structural and biochemical understanding of hMOB-regulated signalling in order to fully elucidate the molecular mechanisms underlying human diseases, such as cancer, that are linked to the deregulation of hMOBs. In this regard, research into the cellular functions of MOBs is also timely and relevant. The biological roles of MOBs must likewise be addressed in appropriate animal models, such as transgenic mice. To date, only MOB1 knock-out animals have been reported. To our knowledge, transgenic mice allowing for studies of the physiological functions of MOB2, MOB3s and MOB4 are currently not available. Hopefully studies of disease-relevant cell biological roles of MOBs will soon be complemented by appropriate loss-of-function animal models. Many research avenues must be pursued to adequately describe the complexity of MOB signalling in multicellular organisms. Undoubtedly, exciting new discoveries related to MOBs are to be expected over the coming years.

## Figures and Tables

**Figure 1 cells-08-00569-f001:**
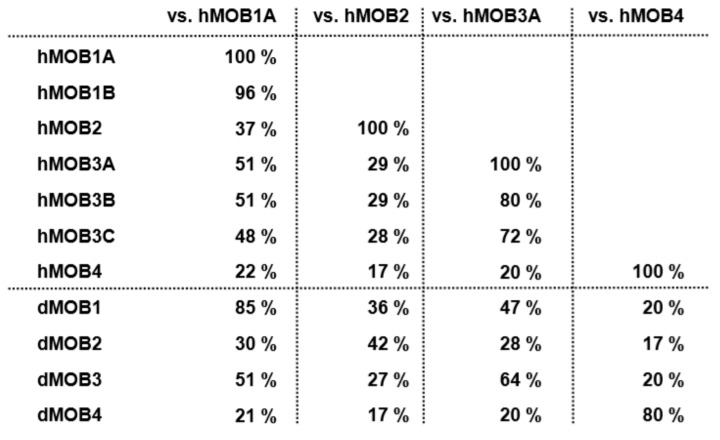
Primary sequence identities of the fly and human MOBs. The identities between primary sequences are displayed. Identities were defined using EMBOSS needle for pairwise alignments (https://www.ebi.ac.uk/Tools/psa/emboss_needle/). The UniProtKB nomenclature for hMOBs can be found in the introduction section of the main text. The UniProtKB names for dMOBs are as follows [4]: dMOB1 (aka Mats and CG13852)–Q95RA8, dMOB2 (aka CG11711)–Q8IQG1, dMOB3 (aka CG4946)–Q9VL13, and dMOB4 (aka CG3403)–Q7K0E3.

**Figure 2 cells-08-00569-f002:**
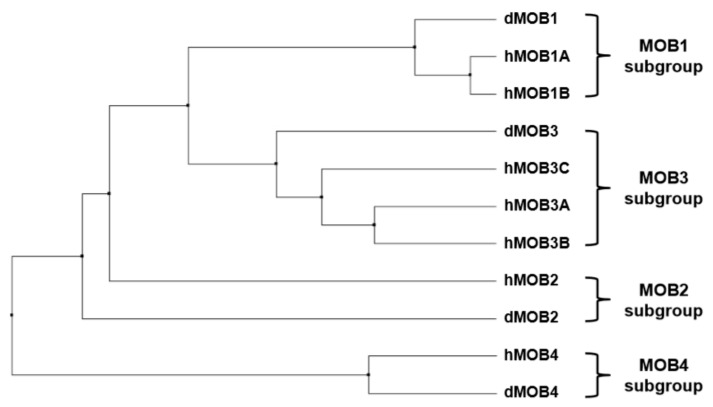
Phylogenetic relationships within the fly and human MOB protein family. The phylogenetic tree was defined using Clustal Omega (https://www.ebi.ac.uk/Tools/msa/clustalo/) together with the Jalview 2.10.5 software using phylogenetic calculation based on the neighbour-joining method. The UniProtKB nomenclature for the analysed proteins is defined in the legend of Figure 1.

**Figure 3 cells-08-00569-f003:**
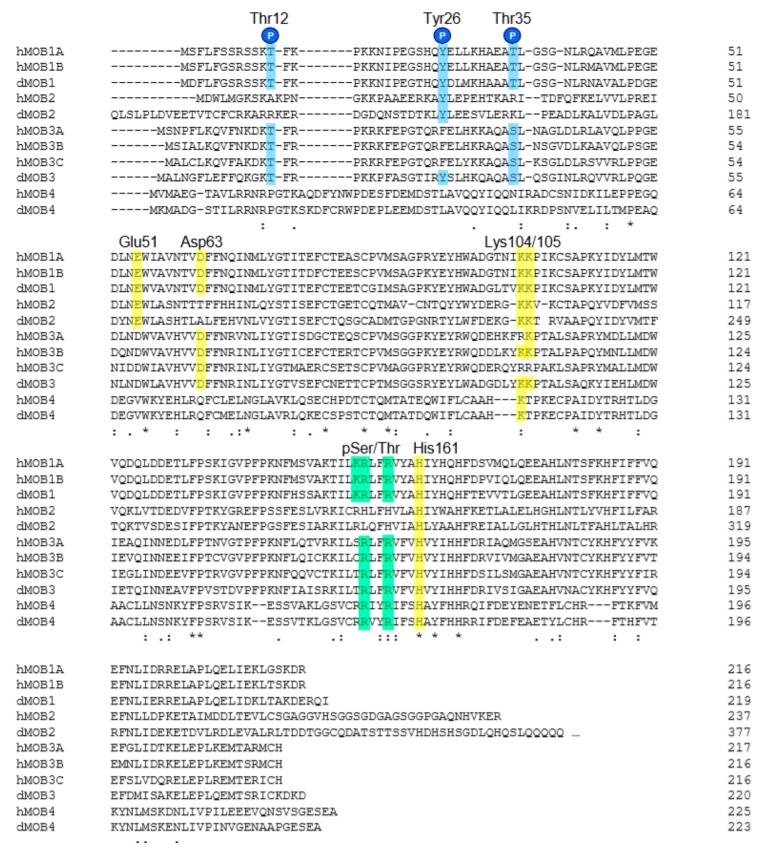
Comparison of the primary protein sequences of human and fly MOBs. Primary protein sequences were aligned using Clustal Omega as defined in the legend of Figure 2. The sites of Thr12, Tyr26 and Thr35 phosphorylations of hMOB1A are highlighted in light blue. Important interaction sites of hMOB1A that were verified by different experimental approaches are displayed in yellow. The key residues of the phospho-Ser/Thr (pSer/Thr) binding pocket of hMOB1A are shown in green. For more details on key residues of hMOB1 that are involved in kinase binding please consult refs. [3,6,70,71,72,78,92,95]. The UniProtKB nomenclature for the analysed proteins is defined in the legends of Figure 1 and Figure 2.

**Figure 4 cells-08-00569-f004:**
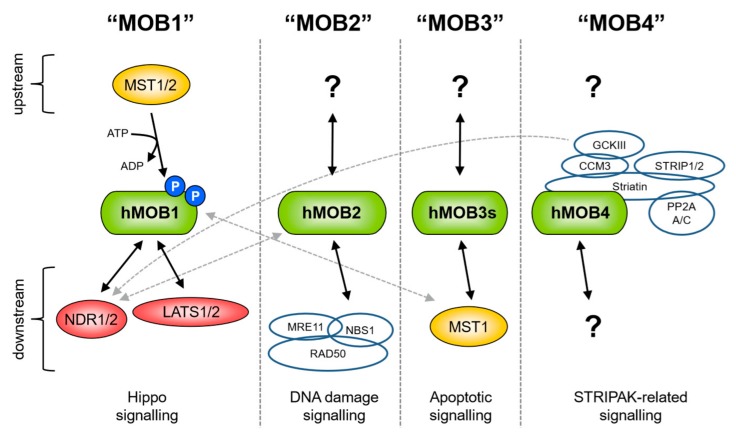
The four branches of MOB control. The “MOB1 branch”: In human cells, hMOB1 phosphorylated by MST1/2 promotes the activation of NDR1/2 and LATS1/2 in the context of Hippo signalling and the regulation of other important cellular functions [1,55,57]. The “MOB2 branch”: hMOB2 can interact with the MRN (Mre11-Rad50-Nbs1) DNA damage sensor complex, thereby supporting DNA damage and consequently cell cycle signalling [37,38]. The “MOB3 branch”: hMOB3s form a complex with MST1, in so doing regulating apoptotic signalling [39]. The “MOB4 branch”: hMOB4 as part of the STRIPAK (Striatin-interacting phosphatase and kinase) complex is likely to support the diverse cellular roles of the STRIPAK complex [43,44]. Notably, it is possible that these MOB branches do not signal in isolation, but rather may be interlinked as highlighted by dashed grey lines. For example, hMOB2 may connect with the “MOB1 branch” by competing with hMOB1 for NDR1/2 binding [35,62], the hMOB3s may link with the “MOB1 branch” through competing with hMOB1 for MST1/2 binding [6,39], or hMOB4 as part of the GCKIII-containing STRIPAK complex may connect with the “MOB1 branch” through GCKIII-mediated phosphorylation of NDR1/2 (see subchapter 4). Noteworthy, Hippo and STRIPAK signalling have already been found to be interconnected (see subchapter 4), although, to our knowledge, the specific roles of hMOB4 have not been defined. hMOBs are in green. MST1/2 and NDR/LATS kinases are in yellow and red, respectively. Established interactions are highlighted by black lines, while putative interconnections between “MOB branches” are shown by dashed grey lines. Phosphorylations are indicated by “P” in blue.

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
