# Peer review of "MOB (Mps one Binder) Proteins in the Hippo Pathway and Cancer"

_cells, 2019, doi:10.3390/cells8060569_

Round 1
Reviewer 1 Report
Authors review in a very extensive way MOB protein family, in a variety of species and in different context such as unicellular and multicellular organisms, development, morphogenesis and disease, among many others.
MOB proteins, as the authors state, have a very pleiotropic activities and one important is to function as binding-regulating proteins in a variety of signaling pathways. The authors revised many of these pathways in different models. The title of the review seems to induce to think that Hippo pathway and cancer would be the main focus of the discussion. However, chapter 8 is very short, descriptive, with little discussion and no conclusive remarks are presented or proposed.
Probably one of the weakness of the review is that tries to include almost everything what is reported about MOBs and does not focus in a more specific subject (cancer for example).
Even though is a valuable review since summarized most of recent literature on this protein family.
Author Response
We thank reviewer#1 for acknowledging that we are providing a very extensive and valuable review of the MOB protein family. Given that MOB-regulated signaling is only an emerging research field we have opted to provide a broad overview of the different intracellular mechanisms and possible links to diseases. This will empower the reader to properly evaluate the roles of MOB proteins in the Hippo pathway and disease such as cancer.
With regard to the specific reviewer’s remark regarding chapter 8, it is imperative to note that chapter 8, composed of around 1,400 words, is not standing alone regarding the discussion of MOBs in cancer. Chapter 7, composed of around 1,400 words, is also fully dedicated to discussing the cancer-associated cellular functions of MOBs. Furthermore, chapters 5 and 6 are also covering some specific aspects of protein-protein interactions and post-translational modifications in the context of cancer. Thus, the topic “cancer and MOBs” is covered by about one third of this extensive review. Of note, the topic “MOBs and the Hippo pathway” is the main focus of chapters 4, 5 and 6, which together comprise nearly 4,000 words, hence this second main topic is also the pure focus of more than one third of this review. Consequently, we strongly feel that the title of this review is justified by its content.
As a second specific point, reviewer#1 implies that the breadth of our coverage is a weakness. We disagree with this conclusion, since the focus of our review is on MOBs and the Hippo pathway and cancer, as already illustrated above. In our opinion, the additional chapters on MOBs in flies, human cells and diseases other than cancer (i.e. chapters 2, 3 and 9) are crucial to properly equip the expert as well as the non-expert with knowledge that is very likely to be important for the design and interpretation of future studies of MOBs in the context of the Hippo pathway and cancer.
Reviewer 2 Report
The manuscript, entitled "MOB (Mps One Binder) Proteins in the Hippo Pathway and Cancer” submitted by Ramazan Gundogdu and Alexander Hergovich tried to summarize the cellular control of MOBs in different organisms and their role in different diseases.
The comments for this manuscript are as follows-
1. Is there any MOBs inhibitor or activator known so far and have been used in a clinical trial?
2. Any available report on the role of MOBs in Acute myeloid leukemia and infertility?
If the above information is available, please include it.
Author Response
We thank reviewer#2 for their feedback.
Regarding the specific comments of reviewer#2 we have the following responses:
1) To our knowledge neither inhibitors nor activators of MOBs have been used in clinical trials, hence we did not add any additional information to the manuscript in this regard.
2) We agree with the reviewer that reports linking MOBs to AML and/or (in-)fertility would be very interesting to include in this review. However, we are not aware of any reports that directly link MOBs to AML or infertility. Consequently, we also did not include any additional information to the manuscript in this regard.
Reviewer 3 Report
The authors summarized the recent progress of MOB scaffold proteins in the Hippo signaling pathway. Especially they are well described role for the phosphorylation-dependent regulation of the NDR/LATS family kinases. The review emphasizes the need to present new perspective and future directions from the author's group since they have much experience in MOB-dependent signaling.The simplified nomenclature of human MOB proteins is much needed in the field to avoid confusion. Therefore, I recommend publication with minor revision, and the work provides a nice summary of current understanding on the roles of MOBs as central signaling adaptors in diverse cellular functions and cancer.
Minor points:
p.10, line 413: The notion that pT378 of MST2 is not the only phospho-Thr site recognized by hMOB1 was first demonstrated in Ref. 72 then later acknowledged and confirmed in Refs. 78 & 92. The multiple phospho-Thr sites in the MST linker is well-described and established in Ref 72. This work should be clearly stated and referenced here.
p.10, line 416: Refs 78 & 92 were wrongly cited here. The importance of the K104/K105/P106 binding motif can be easily explained by the crystal structure of pT378-MST2–MOB1 in Ref 72 since these three residues are within 4-5 Å distance to pT378 residue. Their positive charges are likely to contribute favorably to pMST2 binding.
p.11, line 424: change SLMA to SLMAP; add Ref 68 here since this is the First paper to show that SLMAP-FHA domain binds to multiple phospho-Thr sites in the MST2 linker. The crystal structure of SLMAP-FHA–pMST2 nicely explained the binding specificity. This work should be described here.
p.11, line 427: add Ref 68.
p.11, line 431: the effect of LATS1 S690 on MOB1 binding is clearly demonstrated by pMST2–LATS1 structure in Ref. 72. The authors should cite the reference and describe the work more carefully here. There is more understanding on the LATS–pMOB1 interaction than what is stated here. The current description lacks significant insights.
Author Response
We thank reviewer#3 very much for their kind words and their very constructive comments that have helped us to further improve this manuscript. We have tried our best to adequately address the five minor points raised by reviewer#3:
1) Page 10, line 413
As requested, we are citing reference 72 also in the highlighted sentence.
2) Page 10, line 416
As requested, reference 72 was added in this context to references 78 and 92.
3) Page 11, line 424
As requested, we changed SLMA to SLMAP and added reference 68 in this context.
4) Page 11, line 427
As requested, reference 68 was added to reference 69.
5) Page 11, line 431
As requested, also this minor point was fully addressed by re-phrasing the text section at question and further including a brief discussion of reference 72 in the context of LATS1 binding to phospho-MOB1.